# Prevalence and regional disparities of undiagnosed diabetes mellitus in Bangladesh: Results from the Bangladesh Demographic and Health Survey data

Ahmed Hossain[1,2]*, Shakib Ahmed Suhel[2], Shofiqul Islam[2], Nipa Rani Dhor[2], Nayma Akther[2], Shubrandu Sutradhar Sanjoy[3], Saifur Rahman Chowdhury[4]

1 College of Health Sciences, University of Sharjah, Sharjah, United Arab Emirates, 2 Department of Public Health, North South University, Dhaka, Bangladesh, 3 Saskatchewan Health Authority, Regina, Saskatchewan, Canada, 4 Department of Health Research Methods, Evidence, and Impact, McMaster University, Hamilton, Ontario, Canada

* ahmed.hossain@northsouth.edu

## Abstract

### Background

While undiagnosed diabetes mellitus (DM) presents a substantial global concern, there is a dearth of research examining its prevalence and characteristics specifically within the regional context of Bangladesh. The study focused on assessing the prevalence of undiagnosed diabetes mellitus in Bangladesh and examining regional disparities.

### Methods

The study analyzed data from the Bangladesh Demographic and Health Survey conducted between 2017 and 2018. The analysis focused on 11,911 participants aged 18 and above. Prevalence rates of both diagnosed and undiagnosed DM were calculated across various demographic and regional factors. To understand the impact of socio-demographic and regional variables on diagnosed and undiagnosed DM, the study employed multinomial regression analysis.

### Results

The study encompassed 11,911 participants with an average age of 39, of whom 57% were females. Among them, 333 individuals (2.8%) were diagnosed with diabetes mellitus (DM), while 667 participants (5.6%) had undiagnosed DM. The prevalence of both diagnosed and undiagnosed DM was notably higher in elderly, hypertensive, overweight or obese, and rural residents. Regression analysis indicated that individuals aged 70 and above faced 2.14 times more likely of diagnosed diabetes compared to those aged 30-39 (RRR = 2.20; 95% CI = 1.35-3.58). Regarding residential regions, individuals from the city exhibited significantly higher prevalence rates for both diagnosed DM (RRR: 1.83; 95% CI = 1.31-2.57) and undiagnosed DM (RRR: 1.52; 95% CI = 1.18-1.95) compared to those from the rural of Bangladesh.

**Data availability statement:** The study used data from the Bangladesh Demographic and Health Survey 2017-18. The data sets are available at: https://dhsprogram.com/data/available-datasets.cfm.

**Funding:** The author(s) received no specific funding for this work.

**Competing interests:** The authors have declared that no competing interests exist.

## Conclusion

The high prevalence of undiagnosed DM in city areas suggests potential shortcomings in routine diabetes screening practices. Prioritizing screening, particularly for high-risk groups like older adults, individuals with elevated BMI, hypertension, and urban residents from the central region of the country, is crucial. These groups have elevated diabetes risk and face higher complications without timely detection and treatment. To address this issue, collaborative efforts among the Bangladeshi government, healthcare providers, and community organizations are imperative.

## Introduction

Diabetes mellitus has emerged as a critical global health challenge, characterized by chronic hyperglycemia resulting from defects in insulin secretion, insulin action, or both [1,2]. The disease is associated with severe complications, including stroke, kidney failure, myocardial infarction, blindness, and lower limb amputation, among others [1–3]. These complications contribute to the substantial burden that diabetes places on healthcare systems worldwide.

The prevalence of diabetes has been steadily increasing, leading to severe health complications and a substantial burden on healthcare systems worldwide [3]. Alarming trends show an increase in diabetes prevalence, particularly in low- to middle-income countries, where approximately 79% of diagnosed individuals reside [4]. Projections indicate that Southeast Asia will see a 74% rise in diabetes cases by 2045, escalating from 88 million in 2019 to an estimated 153 million [5]. Another study indicates that the prevalence of diabetes would rise from 9% between 2006 and 2010 to 13% in 2030 [6]. According to WHO, 3% of total deaths are due to diabetes, and 12.88 million people are affected by diabetes [7]. An early diagnosis or screening is necessary for a country since undiagnosed diabetes is linked to potentially avoidable and expensive diabetes complications [7–9].

Understanding the demographic and health-related factors associated with diabetes and undiagnosed diabetes is crucial for developing effective prevention and management strategies [9–11]. A variety of factors, including age, family history, hypertension, obesity, physical inactivity, and smoking, are linked to the onset of diabetes [11–13]. Furthermore, sleep disturbances and insufficient public health measures, such as ineffective smoking bans, may contribute to the disease's rise [13–16]. Identifying these risk factors and understanding regional disparities in diabetes prevalence is essential for policymakers to create effective prevention and management strategies to reduce the disease's burden.

In Bangladesh, diabetes presents a significant public health concern. Recent national surveys indicate an estimated prevalence of 8.1%, with over 10% of the population potentially unaware of their condition [17,18]. This unawareness can lead to avoidable complications and underscores the importance of early diagnosis and screening.

Bangladesh has one of the highest rates of diabetes worldwide, yet a significant number of cases remain undiagnosed [19]. Identifying regional disparities is crucial for targeting areas that require the most urgent intervention. Rural and underserved regions often face challenges such as limited healthcare facilities, inadequate diagnostic services, and a shortage of trained professionals. Analyzing regional differences can shed light on healthcare gaps and pave the way for improving accessibility. Additionally, understanding these variations can aid in designing effective early detection programs and preventive measures. By leveraging regional data, government and healthcare policymakers can allocate resources more efficiently, ensuring that high-priority areas receive the necessary attention for diabetes screening

and management. Addressing regional disparities in undiagnosed diabetes in Bangladesh is essential for developing targeted healthcare strategies, enhancing early diagnosis, and mitigating long-term health and economic impacts.

The study focused on assessing the prevalence of undiagnosed diabetes mellitus in Bangladesh and examining regional disparities. The research used data from the Bangladesh Demographic and Health Survey. The findings shed light on the extent of undiagnosed diabetes across different regions of the country, offering valuable insights into the hidden burden of this condition.

## Materials and methods

### Study design and setting

The Bangladesh Demographic and Health Survey (BDHS) is a comprehensive national survey that gathers extensive data on various health and demographic indicators within Bangladesh [20]. The National Institute of Population Research and Training (NIPORT), under the Ministry of Health and Family Welfare, oversees the periodic conduct of this survey. The data collected is essential for informing health policy decisions and program implementation in Bangladesh. The BDHS is part of the global Demographic and Health Surveys (DHS) program, which is primarily funded by the United States Agency for International Development (USAID) and receives technical support from ICF International. Through its systematic collection of data, the BDHS provides valuable insights that contribute to understanding health trends and challenges in the country.

The 2017-2018 BDHS, a cross-sectional study, covered urban and rural areas using a sampling frame from the 2011 census. The participant selection process for this study is shown in Fig 1. A total of 672 primary sampling units (PSUs) were selected, consisting of approximately

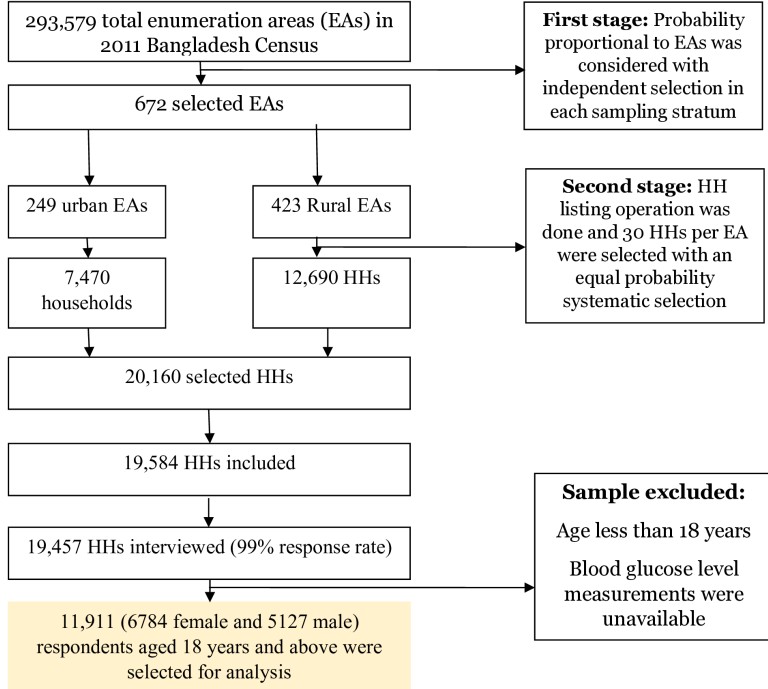

**Fig 1. Flowchart of participation selection for the analysis.**

120 households each, resulting in a final participant count of 11,911 adults after excluding individuals under 18 years and those with incomplete diabetes data.

## Outcome measure

The fasting blood glucose level of the participants was used for measuring diabetes and undiagnosed diabetes. The BDHS used HemoCue Glucose 201 + blood glucose analyzers system to collect blood from the capillary from the middle or ring finger after they had fasted overnight [20]. According to the American Diabetic Association and WHO classifications, people with fasting who have blood glucose ≥ 7.0 mmol/L (126 mg/dl) consider diabetic participants [21,22]. We defined diabetes according to WHO classification and taking into consideration of medication use for T2DM. In addition, our study considers those diagnosed with diabetes who have fasting glucose level was 7.0 mmol/L or more, those who took prescribed medicine, or those who were told by health professionals (doctor/nurse) that they had diabetes. However, undiagnosed diabetes was considered that participant FPG was ≥ 7.0 mmol/L, they had never taken prescribed medicine, and they had never been told that they had diabetes by doctors or nurses. Thus, the analysis categorizes participants into three groups: those with normal glucose levels, those diagnosed with diabetes, and those with undiagnosed diabetes.

## Independent variables

In our study, we included variables at the individual, household, and community levels. Variables were categorized by age in years (< 30, 30–39, 40–49, 50–59, 60–69, and ≥70), sex (Male, and Female), marital status (Never married, Married, Widowed/Divorced/Separated), education (No schooling, Primary, Secondary, and Higher-secondary and above), body mass index (BMI) (Underweight: BMI < 18.5 kg/m2, Normal weight: BMI 18.5–24.99 kg/m2, Overweight: BMI 25–29.99 kg/m2, and Obese: BMI ≥30 kg/m2). High Blood Pressure, SBP ≥140 mmHg and/or DBP ≥90 mmHg and/or currently on treatment with antihypertensive medication was considered as a hypertensive patient.

In the BDHS, wealth status is assessed using a wealth index that classifies households into quintiles based on assets and amenities, such as ownership of consumer goods and housing characteristics [20]. This index is calculated through principal component analysis (PCA) to rank households into five categories: Poorest, Poorer, Middle, Richer, and Richest. For analysis, the poorest and poorer categories are combined as "poor," while the richer and richest are grouped as "rich," resulting in three overall wealth status categories.

The residential region is categorized into city, semi-urban, and rural areas, while the geographic region of residence within the country is classified according to its administrative divisions. We categorized Barisal, Chattogram, and Khulna to be in the coastal region, Dhaka and Mymensingh in the central region, Rajshahi and Rangpur in the North region, and Sylhet in the East region.

## Data analysis

The analysis was conducted using R version 4.3.0. Survey weights from the BDHS were applied to adjust the statistics to ensure representation of the broader population. Continuous variables were expressed as mean and standard deviation, while categorical data were presented as numbers and percentages. Baseline characteristics were summarized for normal, diagnosed, and undiagnosed diabetes groups, as well as their subgroups.

A multinomial regression model was employed to analyze related variables, with three outcome categories representing participants with normal glucose level, diagnosed, and undiagnosed diabetes. The "normal glucose level" category was used as a reference point against

which the other two were compared. Relative risk ratios (RRR) with 95% confidence intervals (CI) were calculated from the multinomial regression coefficients, with the results exponentiated. The RRR, derived from a multinomial logistic regression model, quantifies how the risk of developing diabetes and undiagnosed diabetes in a comparison group changes relative to the normal glucose level group with a one-unit change in the predictor variable, while keeping all other variables constant. To enhance our primary analysis, we conducted a sensitivity analysis using a machine learning approach, specifically the Random Forest (RF) method. We also applied the multilevel multinomial logistic regression model in the sensitivity analysis. Statistical significance in this study was defined as a p-value of 0.05 or lower.

## Ethics statement

The study used deidentified data from the Demographic Health Survey program, which has already received ethical approval from the participating countries, no further ethical permission was sought to carry out this research. Data was collected from online source (https://dhsprogram.com) with appropriate request. Written informed consent from the respondents enrolled in the survey and other ethical review documents are available at: https://dhsprogram.com/methodology/Protecting-the-Privacy-of-DHS-Survey-Respondents.cfm.

## Results

### Characteristics of the study participants

Table 1 provides insights into the prevalence of normal, diabetic, and undiagnosed diabetes cases among 11,911 participants, along with their socio-demographic characteristics. The participants' average age was 39, with 33% below 30 years and 24% aged 30-39. Around 6% were aged 70 or above, and 43% were male. Education-wise, 25% were illiterate, 30% had primary education, 29% had secondary education, and 16% had higher education. Additionally, 64% lived in rural areas, 27% in semi-urban areas. The table shows that 17% were underweight, 20% were overweight, and 23% had high blood pressure. In terms of wealth index, 39% were from the poor or poorest families, while 42% were from rich or the richest families. Most participants (80%) were married and 39% were not engaged in physical labor. Geographically, 38% were from coastal regions (Barishal, Chittagong, and Khulna), 24% from central regions (Dhaka and Mymensingh), and 26% from northern regions (Rajshahi and Rangpur) of Bangladesh.

### Prevalence of diagnosed and undiagnosed diabetes mellitus

The study examined Diabetes Mellitus prevalence in participants, revealing that 2.8% had diagnosed diabetes, and 5.6% had undiagnosed diabetes. The results in Table 1 show that diabetes prevalence showed slight variations by gender, with females having a higher rate of diagnosed diabetes (2.9%) compared to males (2.7%), while males had a higher prevalence of undiagnosed diabetes (5.9%) than females (5.4%). The overall prevalence of diabetes (both diagnosed and undiagnosed) rose significantly with age (p < 0.001). The youngest age group (<30 years) had the lowest rates of diagnosed (0.02%) and undiagnosed diabetes (3.8%). The highest prevalence of undiagnosed diabetes was observed in the 50–59 age group (7.4%), while diagnosed diabetes peaked at 6.6% among those aged 60–69. Married individuals, who made up 80.2% of the sample, exhibited higher rates of both diagnosed (3.0%) and undiagnosed diabetes (5.8%) compared to those who were never married or widowed/divorced (p < 0.001). Additionally, diabetes prevalence decreased with higher education levels (p = 0.004).

**Table 1. Distribution of healthy, diagnosed diabetes mellitus and undiagnosed diabetes mellitus people by socio-demographic groups of the respondents.**

| Variables | Normal glucose level (n = 10911, 91.6%) | Diagnosed diabetes (n = 333, 2.8%) | Undiagnosed diabetes (n = 667, 5.6%) | Total = 11911 | P-value |
|---|---|---|---|---|---|
| **Sex** | | | | | |
| Female | 6222 (91.7%) | 195 (2.9%) | 367 (5.4%) | 6784 (57.0%) | 0.499 |
| Male | 4689 (91.5%) | 138 (2.7%) | 300 (5.9%) | 5127 (43.0%) | |
| **Age group** | | | | | |
| 30-39 | 2612 (91.9%) | 60 (2.1%) | 171 (6.0%) | 2843 (23.9%) | <0.001 |
| <30 | 3751 (95.9%) | 9 (.02%) | 150 (3.8%) | 3910 (32.8%) | |
| 40-49 | 1825 (89.4%) | 82 (4.0%) | 135 (6.6%) | 2042 (17.1%) | |
| 50-59 | 1152 (86.2%) | 85 (6.4%) | 99 (7.4%) | 1336 (11.2%) | |
| 60-69 | 945 (87.7%) | 71 (6.6%) | 61 (5.7%) | 1077 (9.0%) | |
| >=70 | 626 (89.0%) | 26 (3.7%) | 51 (7.3%) | 703 (5.9%) | |
| **Marital Status** | | | | | |
| Married | 8716 (91.3%) | 284 (3.0%) | 550 (5.8%) | 9550 (80.2%) | <0.001 |
| Never married | 1182 (95.7%) | 2 (.02%) | 51 (4.1%) | 1235 (10.4%) | |
| Widowed/divorced | 1013 (90.0%) | 47 (4.2%) | 66 (5.9%) | 1126 (9.5%) | |
| **Education** | | | | | |
| No education | 2731 (92.1%) | 72 (2.4%) | 163 (5.5%) | 2966 (24.9%) | 0.004 |
| Primary | 3321 (91.5%) | 80 (2.2%) | 227 (6.3%) | 3628 (30.5%) | |
| Secondary | 3143 (91.1%) | 123 (3.6%) | 184 (5.3%) | 3450 (29.0%) | |
| College or higher | 1716 (91.9%) | 58 (3.1%) | 93 (5.0%) | 1867 (15.7%) | |
| **Working status** | | | | | |
| Yes | 6715 (92.5%) | 172 (2.4%) | 374 (5.2%) | 7261 (61.0%) | <0.001 |
| No | 4196 (90.2%) | 161 (3.5%) | 293 (6.3%) | 4650 (39.0%) | |
| **Wealth status** | | | | | |
| Poor | 4371 (95.0%) | 39 (.08%) | 192 (4.2%) | 4602 (38.6%) | <0.001 |
| Middle | 2198 (93.2%) | 46 (1.9%) | 115 (4.9%) | 2359 (19.8%) | |
| Rich | 4342 (87.7%) | 248 (5.0%) | 360 (7.3%) | 4950 (41.6%) | |
| **Hypertension** | | | | | |
| No | 8508 (93.3%) | 161 (1.8%) | 451 (4.9%) | 9120 (76.6%) | <0.001 |
| Yes | 2403 (86.1%) | 172 (6.2%) | 216 (7.7%) | 2791 (23.4%) | |
| **BMI groups** | | | | | |
| Normal | 6467 (92.7%) | 166 (2.4%) | 342 (4.9%) | 6975 (58.6%) | <0.001 |
| Underweight | 1942 (95.3%) | 10 (0.05%) | 85 (4.2%) | 2037 (17.1%) | |
| Overweight | 2084 (86.8%) | 123 (5.1%) | 194 (8.1%) | 2401 (20.2%) | |
| Obese | 418 (83.9%) | 34 (6.8%) | 46 (9.2%) | 498 (4.2%) | |
| **Residential regions** | | | | | |
| City | 944 (87.6%) | 50 (4.6%) | 84 (7.8%) | 1078 (9.1%) | <0.001 |
| Semi-urban | 2874 (90.3%) | 108 (3.4%) | 202 (6.3%) | 3184 (16.7%) | |
| Rural | 7093 (92.7%) | 175 (2.3%) | 381 (5.0%) | 7649 (64.2%) | |
| **Geographic regions** | | | | | |
| North | 2938 (93.6%) | 85 (2.7%) | 116 (3.7%) | 3139 (26.4%) | <0.001 |
| Central | 2553 (89.7%) | 69 (2.4%) | 223 (7.8%) | 2845 (23.9%) | |
| Coastal | 4122 (91.3%) | 148 (3.3%) | 244 (5.4%) | 4514 (37.9%) | |
| East | 1298 (91.9%) | 31 (2.2%) | 84 (5.9%) | 1413 (11.9%) | |

Individuals without formal education had higher rates of diagnosed diabetes (2.4%) compared to those with college or higher education (3.1%). Those who were unemployed showed a higher prevalence of both diagnosed (3.5%) and undiagnosed diabetes (6.3%) compared to employed individuals (p < 0.001). Diabetes prevalence also varied by wealth status, with a significant inverse relationship observed (p < 0.001). Poorer individuals had lower rates of diagnosed diabetes (0.08%) and undiagnosed diabetes (4.2%), while wealthier individuals had the highest prevalence of both diagnosed (5.0%) and undiagnosed diabetes (7.3%). Hypertension was strongly associated with higher rates of both diagnosed (6.2%) and undiagnosed diabetes (7.7%) compared to those without hypertension (p < 0.001). Obesity was linked to the highest rates of both diagnosed (6.8%) and undiagnosed diabetes (9.2%), whereas underweight individuals had the lowest rates (p < 0.001).

Geographically, semi-urban residents had higher rates of both diagnosed and undiagnosed diabetes compared to rural residents (p < 0.001). City dwellers exhibited the highest prevalence of undiagnosed diabetes at 7.8%. Regional differences were also significant (p < 0.001), with the central region showing the highest prevalence of undiagnosed diabetes at 7.8%. The findings indicate that older age groups, obesity, and hypertension were strongly associated with higher rates of both diagnosed and undiagnosed diabetes. Higher wealth status and urban or city residence were linked to increased prevalence, while higher education levels appeared to have a protective effect.

## Age-specific prevalence of diabetes across residential regions

The Fig 2 illustrates the age-specific prevalence of diagnosed and undiagnosed diabetes across different residence types (City, Rural, and Semi-Urban). The prevalence of diabetes generally rises with age, reaching its highest point in the 60-69 age group, particularly among city residents. City and semi-urban areas show a higher prevalence of diagnosed diabetes across most age groups compared to rural regions. This trend suggests that urban residents may benefit from better access to diabetes screening and healthcare services, resulting in higher diagnosis rates. In contrast, the prevalence of undiagnosed diabetes does not follow a consistent upward trend with age. Younger age groups (18-29, 30-39) exhibit a notably higher proportion of undiagnosed cases compared to diagnosed ones. Rural and semi-urban areas tend to have higher rates of undiagnosed diabetes, highlighting potential gaps in healthcare access and screening services in these regions.

## Associated factors: multinomial regression analysis

The analysis examines the relative likelihood of diagnosed and undiagnosed diabetes mellitus compared to individuals with normal glucose levels, using various demographic, health, and socioeconomic factors. We previously utilized a directed acyclic graph (DAG) – a visual representation of hypothesized causal relationships – to identify confounders in these relationships and it is shown in supporting information S3 Fig in S1 File. In this DAG, diabetes status was considered the outcome, while residency was the main exposure. We also investigated the covariates to identify the confounding effects in relation to residential regions (See supporting information s2 Fig in S1 File). For the association between exposures and diabetes, a minimal sufficient adjustment set including age, sex, BMI, hypertension and residency was determined. The AIC values of the model is given in the supporting information S2 Table in S1 File. The Table 2 provides the results from the multinomial logistic regression model.

Males showed a slightly lower likelihood of diagnosed diabetes compared to females (RRR = 0.88, 95% CI: 0.70–1.12), though this difference was not statistically significant. Males also exhibited a higher likelihood of undiagnosed diabetes (RRR = 1.11, 95% CI: 0.95–1.31), but

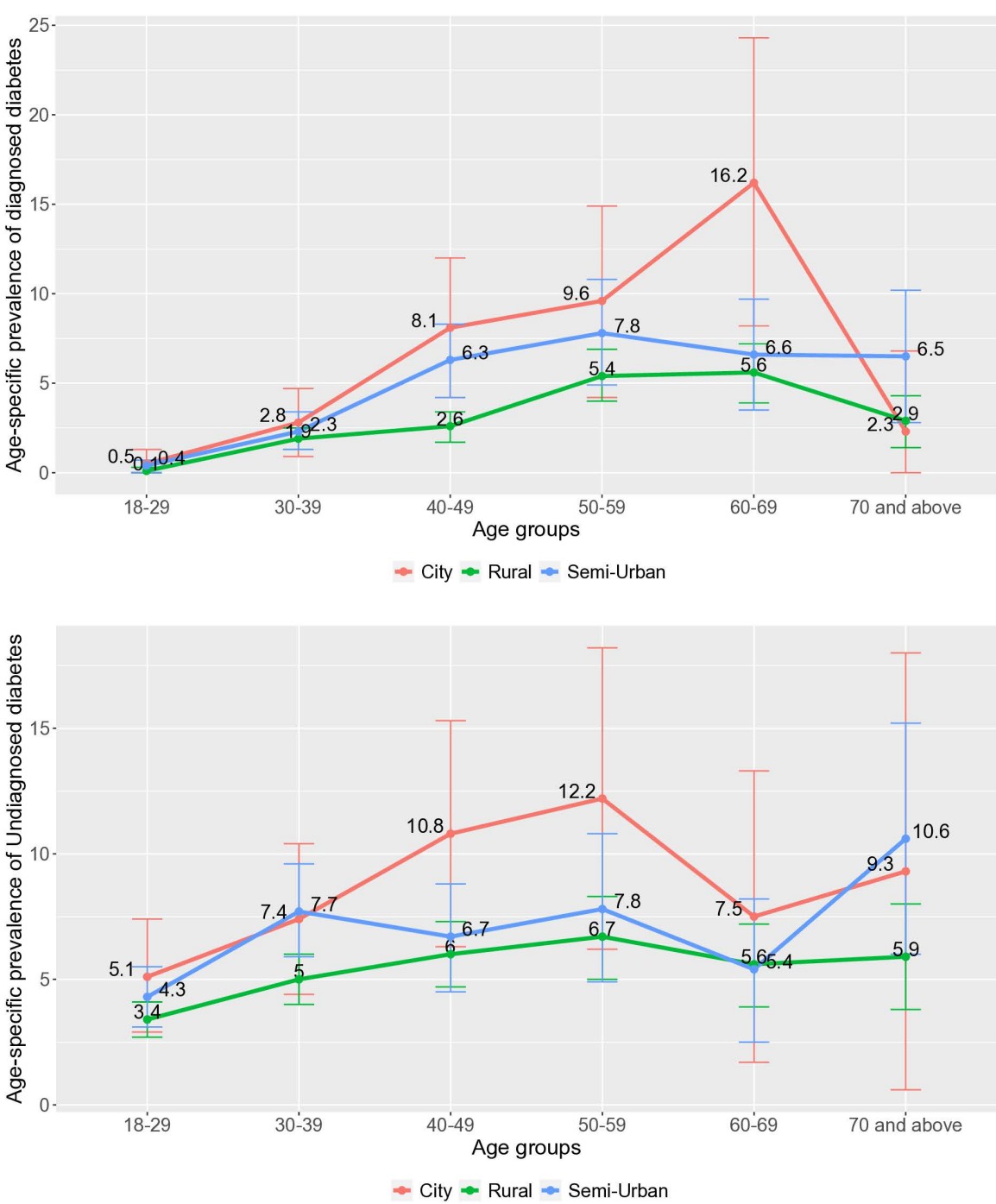

**Fig 2. Age-specific prevalence of diagnosed and undiagnosed diabetes across different residence types.**

this was similarly not statistically significant. The absence of strong associations suggests that sex alone may not be a significant determinant of diabetes likelihood without considering other factors such as BMI or lifestyle.

The likelihood of diabetes increased significantly with age, reaching its peak in the 60–69 age group (RRR = 3.71, 95% CI: 2.57–5.36) compared to the reference group (30–39 years).

**Table 2. Associated factors of diagnosed diabetes mellitus and undiagnosed diabetes mellitus using multivariable multinomial regression models.**

| Variables | Diagnosed Diabetes mellitus vs. normal | | | Undiagnosed Diabetes mellitus vs. normal | | |
|---|---|---|---|---|---|---|
| | RRR | 95%CI | | RRR | 95%CI | |
| **Sex (ref: Female)** | | | | | | |
| Male | 0.88 | 0.70 | 1.12 | 1.11 | 0.95 | 1.31 |
| **Age group (ref: 30–39)** | | | | | | |
| <30 | **0.14** | **0.07** | **0.28** | **0.70** | **0.56** | **0.88** |
| 40-49 | **1.92** | **1.36** | **2.70** | 1.13 | 0.89 | 1.43 |
| 50-59 | **3.45** | **2.44** | **4.88** | **1.35** | **1.04** | **1.76** |
| 60-69 | **3.71** | **2.57** | **5.36** | 1.04 | 0.76 | 1.42 |
| >=70 | **2.20** | **1.35** | **3.58** | 1.33 | 0.95 | 1.86 |
| **Hypertension (ref: No)** | | | | | | |
| Yes | **1.88** | **1.49** | **2.38** | **1.34** | **1.12** | **1.61** |
| **BMI group (ref: Normal)** | | | | | | |
| Underweight | **0.19** | **0.10** | **0.37** | 0.84 | 0.66 | 1.07 |
| Overweight | **1.96** | **1.52** | **2.51** | **1.63** | **1.35** | **1.97** |
| Obese | **2.44** | **1.62** | **3.65** | **1.89** | **1.35** | **1.97** |
| **Residence (ref: Rural)** | | | | | | |
| Semi-urban | **1.46** | **1.13** | **1.87** | **1.25** | **1.05** | **1.50** |
| City | **1.83** | **1.31** | **2.57** | **1.52** | **1.18** | **1.95** |

Bold faces represent potential risk factor at 5% significance level.

Individuals aged 70 and above also had an elevated likelihood (RRR = 2.20, 95% CI: 1.35–3.58). The highest likelihood for undiagnosed diabetes was observed in the 50–59 age group (RRR = 1.35, 95% CI: 1.04–1.76). Younger individuals (<30 years) had a significantly lower likelihood for both diagnosed (RRR = 0.14, 95% CI: 0.07–0.28) and undiagnosed diabetes (RRR = 0.70, 95% CI: 0.56–0.88). The strong association between older age and diabetes aligns with findings that aging contributes to insulin resistance and beta-cell dysfunction.

Hypertension was associated with nearly double the likelihood of diagnosed diabetes (RRR = 1.88, 95% CI: 1.49–2.38). A significant association was also observed for undiagnosed diabetes (RRR = 1.34, 95% CI: 1.12–1.61). Hypertension is a well-established factor linked to type 2 diabetes due to shared mechanisms such as chronic inflammation and endothelial dysfunction.

Overweight individuals had nearly double the likelihood of diagnosed diabetes (RRR = 1.96, 95% CI: 1.52–2.51), while obese individuals had the highest likelihood (RRR = 2.44, 95% CI: 1.62–3.65). Overweight individuals also showed an elevated likelihood for undiagnosed diabetes (RRR = 1.63, 95% CI: 1.35–1.97), with obesity further increasing this likelihood (RRR = 1.89, 95% CI: 1.35–1.97). Obesity is a major modifiable factor associated with both diagnosed and undiagnosed diabetes due to its role in insulin resistance. Underweight individuals, on the other hand, had a significantly lower likelihood of diagnosed diabetes (RRR = 0.19, CI: 0.10–0.37).

Urban residents had a higher likelihood of diabetes compared to rural residents. City dwellers exhibited the greatest likelihood (RRR = 1.83, CI: 1.31–2.57), followed by semi-urban residents (RRR = 1.46, CI: 1.13–1.87). City residents also had a higher likelihood of undiagnosed diabetes (RRR = 1.52, CI: 1.18–1.95), with semi-urban residents following closely (RRR = 1.25, CI: 1.05–1.50). Urbanization is often associated with lifestyle changes, such as reduced physical activity and increased consumption of processed foods, which contribute to higher diabetes prevalence.

Thus, the results indicates that older age, obesity, and hypertension were strongly associated with a higher likelihood of both diagnosed and undiagnosed diabetes. Urban or city residence and higher wealth status were also linked to increased prevalence, while higher education levels appeared to have a protective effect.

## Sensitivity analysis

We conducted a sensitivity analysis using a multilevel multinomial regression model implemented with the brms package in R. The model applies the geographic residence as the random effect, and results are presented in the supporting information S1 Table in S1 File. The multilevel multinomial logistic regression model was employed to examine regional disparities in the prevalence of diagnosed and undiagnosed diabetes compared to no diabetes. In the results, each parameter is summarized using the mean (Estimate) and standard deviation (Est. Error) of the posterior distribution, along with two-sided 95% Credible Intervals (l-95% CI and u-95% CI) derived from quantiles.

The findings on regional disparities indicate that individuals in city areas are exp(0.60) = 1.822 times more likely to have diabetes compared to those in rural regions. The one-sided 95% credibility interval does not include zero, suggesting statistical significance. Additionally, city areas show that individuals are exp(0.35) = 1.42 times more likely to have undiagnosed diabetes compared to rural regions. Similar results were observed when using a multinomial regression model.

We also used a random forest model to identify the top five important variables in defining diagnosed and undiagnosed diabetes among the adult Bangladeshis in a sensitivity analysis, and results are shown in supporting information S1 Fig in S1 File. The values of mean decrease accuracy estimate the prediction measure of significance. If a particular variable is removed from the model, the higher it is for a predictor, the more the precision will decrease. The overall precision of the test data is 91.7 percent. The findings are listed in the Supplement. The Figure shows that, according to the measure of importance, the top 5 potential predictors are: age, education, residency, BMI group, and presence of hypertension.

## Discussion

The results of the analysis on diagnosed and undiagnosed diabetes mellitus provide critical insights into the demographic and health-related factors influencing diabetes prevalence in the studied population. The research findings indicating that 3% of the Bangladeshi population has received a diabetes mellitus diagnosis, while an additional 6% are undiagnosed, highlight a significant public health concern in Bangladesh. This suggest that approximately 9% of the population is affected by diabetes, either diagnosed or undiagnosed. This aligns with finding from a study of India that estimates have been given by IDF, where diabetes prevalence was estimated at 9.6% in 2021 [23]. Another study from Nepal indicates the prevalence of T2DM in Nepal was 8.5% (95% CI 6.9–10.4%) in 2020 [24]. In 2018, the weighted prevalence of diabetes in the United States was 11.6% (95% confidence interval: 11.0% to 12.1%), which is notably higher than the prevalence of diabetes in Bangladesh [25]. A study conducted in India found that 1.2% of individuals aged 15 to 50 had undiagnosed diabetes [26]. Additionally, the prevalence of undiagnosed diabetes affects 1-2% of US adults which is lower than the undiagnosed rate in Bangladesh [27]. The result emphasizes the importance of continued efforts to increase awareness, improve access to healthcare, and implement effective screening programs to reduce the burden of undiagnosed diabetes.

The analysis suggests that men are slightly more likely to be undiagnosed with diabetes than women, but there is no significant difference in the prevalence of undiagnosed diabetes between the sexes. This may indicate that women are less likely to seek healthcare or have access to medical services, contributing to lower detection rates. A study indicates that the prevalence of type 2 diabetes mellitus is increasing in both sexes, but men are usually diagnosed at a younger age [28]. Understanding these gender dynamics is essential for designing effective outreach programs tailored to both men and women.

We found age is a significant determinant of diabetes prevalence, with older age groups exhibiting higher risks for both diagnosed and undiagnosed diabetes. Individuals aged 50-59 years show a notably high risk for diagnosed diabetes, while those aged 60-69 years have an increased risk for undiagnosed diabetes. A similar trend is observed in Sri Lanka and Indonesia, where the prevalence of diabetes increases with age, reaching its peak among individuals aged 60 to 69 years [29,30]. These findings highlight the importance of age-targeted interventions, particularly for older adults who may require more proactive screening and management strategies to prevent complications associated with diabetes.

The results indicate that education level plays a crucial role in diabetes prevalence. Individuals with primary and secondary education levels have higher risks for diagnosed diabetes compared to those with no education. A study indicates diabetes self-management training and education plays a vital role in the management of diabetes [31]. Interestingly, individuals with college education exhibit a lower risk for diagnosed diabetes but a higher risk for undiagnosed diabetes. This paradox may reflect differences in health literacy, access to healthcare resources, or lifestyle choices among various educational groups.

The study indicates wealth status also significantly impacts diabetes prevalence, with wealthier individuals showing a markedly higher risk for both diagnosed and undiagnosed diabetes. These finding challenges common assumptions that poorer individuals are at greater risk for chronic diseases due to limited access to healthcare and resources. However, a few studies mentioned a similar finding [32,33]. It suggests that wealthier individuals may face sedentary lifestyle-related risks or may be more likely to undergo screening, leading to higher diagnosis rates.

We found hypertension is strongly associated with both diagnosed and undiagnosed diabetes. It is also commonly found in many studies [34–36]. This relationship highlights the need for integrated healthcare approaches that address both hypertension and diabetes management, as they often coexist and exacerbate each other's effects. Body Mass Index (BMI) also plays a significant role in diabetes prevalence. Overweight individuals demonstrate a higher risk for both diagnosed and undiagnosed diabetes, while obese individuals show even greater risks, particularly for undiagnosed cases. These findings emphasize the importance of promoting healthy weight management as part of diabetes prevention strategies.

The prevalence of both diagnosed and undiagnosed diabetes is found high in city areas of Bangladesh. Urban residents in Bangladesh are more likely to be overweight or obese, have hypertension, and belong to higher wealth quintiles - all of which are associated with increased diabetes risk. These factors contribute to the higher prevalence of both diagnosed and undiagnosed diabetes in urban areas. In an article in Bangladesh, the authors found urban regions exhibit higher rates of both diabetes (30%) and prediabetes (31%) compared to rural areas [37]. These disparities are influenced by urbanization, dietary habits, and lifestyle factors, necessitating region-specific interventions. Thus, this regional disparity suggests that local factors, including access to healthcare services, cultural attitudes towards health, and environmental influences, may play critical roles in shaping health outcomes.

## Strength and limitations

The study's strengths include its utilization of a large, nationally representative dataset, enhancing its external validity. The inclusion of clinical factors like blood pressure, body weight, height, and fasting glucose levels adds robustness to the findings. The use of multinomial logistic regression improved the precision of results by addressing the limitations of traditional logistic regression's effect size overestimation.

However, there are certain limitations to consider. Glucose levels were measured using capillary blood, and while a conversion to plasma glucose was applied, this approach may still result in some inaccuracies. Moreover, a single blood glucose measurement to diagnose diabetes in the study can lead to diagnostic errors due to its potential for false positives and false negatives. The study's reliance on cross-sectional data restricts its ability to establish causal relationships. Certain potential confounders like family history of diabetes, lifestyle, diet, physical activity, and medication history were not included, limiting the comprehensiveness of the analysis. Furthermore, reliance on self-reported participant data introduces the potential for response bias, as several variables were based on participants' own accounts. The Events Per Predictor (EPP) rule is a widely accepted guideline used to determine the minimum number of events (observations) needed for each category of outcome variable to ensure the reliability of a multinomial logistic regression model. In this case, the sample size for the smallest outcome category (diagnosed diabetes, with 333 events) falls short of the recommended EPP threshold of 15, which would require at least 345 events.

## Conclusion

The high prevalence of untreated diabetes in city areas remains a significant concern in Bangladesh, contributing to substantial health threats for the population. Overall, the results highlight complex interactions between demographic factors, residential regions, and diabetes prevalence in the studied population. The significant associations found among age, education level, wealth status, hypertension, BMI, and regional disparities underscore the need for tailored public health interventions that address these specific determinants of health. By addressing these factors holistically, we can enhance health outcomes and reduce the burden of diabetes within communities.

## Supporting information

**S1 File.   S1 Table.** Results from a multivariable multilevel logistic regresson model. **S2 Table.** AIC value of Multinomial logistic regression model. **S1 Fig.** Random forest model. **S2 Fig.** Mosaic plot with residenital regions to investigate potential confounders. **S3 Fig.** Dagitty plot to investigate potential confounders.
(PDF)

## Author contributions

**Conceptualization:** Ahmed Hossain, Nayma Akther.

**Data curation:** Ahmed Hossain, Shakib Ahmed Suhel.

**Formal analysis:** Ahmed Hossain, Shakib Ahmed Suhel, Shofiqul Islam, Saifur Rahman Chowdhury.

**Methodology:** Ahmed Hossain, Shakib Ahmed Suhel, Shofiqul Islam, Nipa Rani Dhor.

**Software:** Ahmed Hossain.

**Supervision:** Ahmed Hossain.

**Validation:** Shubrandu Sutradhar Sanjoy.

**Writing – original draft:** Ahmed Hossain, Shakib Ahmed Suhel, Nipa Rani Dhor.

**Writing – review & editing:** Ahmed Hossain, Shakib Ahmed Suhel, Shofiqul Islam, Nayma Akther, Shubrandu Sutradhar Sanjoy, Saifur Rahman Chowdhury.

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
