## [Decision Letter · Decision Letter 0]

1 Oct 2024

PONE-D-23-27830Prevalence and regional disparities of Undiagnosed Diabetes Mellitus in Bangladesh: Results from the Bangladesh Demographic and Health Survey dataPLOS ONE

Dear Dr. Hossain,

Thank you for submitting your manuscript to PLOS ONE. After careful consideration, we feel that it has merit but does not fully meet PLOS ONE’s publication criteria as it currently stands. Therefore, we invite you to submit a revised version of the manuscript that addresses the points raised during the review process.

 Editor comments:

The manuscript has potential but needs a deep review at the moment.The introduction needs to be more concise and focused on diabetes. What was the need to mention hypertension? Or the definition of diabetes? My suggestion is to present an overview of data (for the Southeast Asian countries, then Bangladesh), the lack of knowledge and finally the importance of the current study and its objectives.For methods:It is crucial to give more detailed information on the Bangladesh Demographic and Health Survey (BDHS).What do the authors mean by "We classified the wealth status as a household-level element as Poor, Middle, and Rich groups.". Did you come up with this classification based on what? Is there a reference? If yes, you need to add it. The discussion needs an improvement. The authors explored poorly the previous cited study from 2019 which has found a higher prevalence (was it really higher? can't tell since the authors have not presented CI95%), considering diagnosed diabetes. The BDHS was conducted just one year before. Do the authors discussed the diffference?What are the possiblle interventions you mention in Conclusions that could help reduce diabetes in Bangladesh? This conclusion is very broad. Please provide some examples.

Please submit your revised manuscript by Nov 15 2024 11:59PM If you will need more time than this to complete your revisions, please reply to this message or contact the journal office at plosone@plos.org . Please include the following items when submitting your revised manuscript:

We look forward to receiving your revised manuscript.

Kind regards,

Sheila Rizzato Stopa, PhD

Academic Editor

PLOS ONE

Journal Requirements:

2. Please include a caption for figure 1.

Reviewers' comments:

Reviewer's Responses to Questions

**Comments to the Author**

1. Is the manuscript technically sound, and do the data support the conclusions?

Reviewer #1: Yes

Reviewer #2: Yes

2. Has the statistical analysis been performed appropriately and rigorously? 

Reviewer #1: No

Reviewer #2: Yes

3. Have the authors made all data underlying the findings in their manuscript fully available?

Reviewer #1: No

Reviewer #2: Yes

4. Is the manuscript presented in an intelligible fashion and written in standard English?

Reviewer #1: No

Reviewer #2: Yes

5. Review Comments to the Author

Reviewer #1: Thank you for the invitation to review this manuscript, and I sincerely apologize for the delay in submitting my comments. The paper addresses a crucial issue for Bangladesh—undiagnosed diabetes and its risk factors—by analyzing data from the nationally representative 2017 Bangladesh Demographic and Health Survey. While the paper has merits, I have several observations concerning the paper's significance for Bangladesh, sample selection, analysis, and overall writing. Specific comments are provided below:

1. A similar paper utilizing the same data and variables has already been published by Rakibul M. Islam and colleagues in the Diabetes Research and Clinical Practice Journal (https://www.sciencedirect.com/science/article/pii/S0168822722000407 ). Despite the similarities, the authors did not mention this until the end of the last paragraph in the discussion section, making the newly submitted paper seemingly unnecessary.

2. The authors present several instances of misleading information, and the references used are often misleading and incorrect. For example, please check the third sentence of the second paragraph of the introduction section, where the authors reference a meta-analysis conducted by the International Diabetes Federation. However, all three citations supporting this claim are from individual authors affiliated with Bangladeshi institutions, analyzing BDHS survey data. This pattern of misleading citations is prevalent throughout the manuscript, rendering many claims inaccurate.

3. The considered sample for analysis is not entirely correct. The inclusion of respondents with gestational diabetes in their analysis is misleading and illogical.

4. Authors considered body mass index (BMI) as one of the independent variables in the analysis and used the global classification of BMI recommended by WHO in 2004. However, WHO later updated their classification for the Asian population, including Bangladesh. Since then, WHO recommends using this new classification for studying events related to non-communicable diseases, such as diabetes and hypertension. The authors' analysis in this regard lacks precision. Please see the new classification here: https://pubmed.ncbi.nlm.nih.gov/14726171/#:~:text=The%20cut%2Doff%20point%20for,points%20for%20each%20population%20separately.

5. The segmentation of semi-urban data is unclear and not available in the dataset.

6. The merging of wealth quintile and residential regions is questionable, potentially masking the true scenario for certain groups unnecessarily.

7. The section on the outcome variable is not entirely clear, especially concerning how authors used the multinomial regression model.

8. While the use of a multivariate multinomial model is methodologically acceptable, there is evidence that, for clustered data like BDHS, multilevel analysis is recommended for more precise results. Additionally, it's unclear whether the authors considered sample weight in their analysis. These make findings reported by the authors are less precise.

9. The manuscript is poorly written with inadequate critical review and comparison in the introduction and discussion sections, respectively.

Although I am aware of the PLoS ONE criteria regarding repetition and novelty not being grounds for rejection, considering the need for this paper and other issues, I recommend not accepting it for publication.

Reviewer #2: Dear authors,

Congratulalions for your work!

I consider this to be a high-quality study that uses secondary data from a nationally representative population survey—the Bangladesh Demographic and Health Survey (BDHS)—which evaluated more than 10,000 adults. The research complies with ethical standards and research integrity, and the article adheres to appropriate reporting requirements for data availability.

This is original research, and the experiments and statistical analyses appear to have been conducted to a high technical standard. However, as this is a secondary analysis, I find some of the methodological details from the primary survey to be lacking, or at least a reference where such details can be found. The results from Bangladesh are consistent with findings from other regions, so it is crucial for the authors to highlight what unique contributions this study makes to the understanding of diabetes epidemiology. Please see my comments below.

Abstract

There are two abstracts on the submission. On conclusion, its stated that the groups with elevated risk for diabetes face higher complications without timely detection and treatment. However, I think it´s important to rewrite the sentence to avoid misunderstanding, as the study did not measure the complications.

Introduction

On the 4th paragraph of introduction, you should give a better link between the ideas of hypertension, smoke and sleep. I could not understand why state about sleep directly after citing the hypertension. Please review.

Methods

Considering a nationally representative sample, I think it’s important to describe in details the sample selection process; for example, how the participant was selected in each household? It was only one individual for each household, or all residents? The sample included all age groups? I have doubts about these aspects, as you cited exclusion of individuals below 18 years of age. It would be good to have a guide to see the Figure 1.

The definition of diabetes and undiagnosed diabetes is crucial to this study, as they represent the primary outcomes. Some bias may be introduced when relying on self-reports or prescription data, as these do not necessarily reflect the patient's actual condition. For example, a person may be prescribed diabetes medication but may not adhere to the treatment. It is important to clarify how glycemic levels were measured and to consider these factors as limitations of the study.

Given that the study aims to evaluate regional factors influencing undiagnosed diabetes, I recommend including a detailed description of regional differences, socioeconomic levels, and distinctions between rural and urban areas—perhaps incorporating a map—to enhance the understanding of the results.

On methods (Independent variables), the authors cite that “The residential region of the country is classified based on its administrative divisions” and that was categorized as coastal, central, North, or east region. However, on Table 1, the results are presented as “city”, “semi-urban” or “rural”. Please match the information appropriately.

For wealth status, which parameters were used to classify as poor, middle and rich? Please explain.

Data Analysis: which weights were applied on BDHS. Please describe or cite a reference.

Results

On table 1, on age group, please put the category <30 first. Why the reference category is 30-39 years old?

On “The participants' average age was 39”, it´s important to present the standard-deviation with the average age

Discussion

The discussion section could be improved, exploring the main risk factors related to undiagnosed diabetes, as the main result of the study.

Explaining the structure of the health services could aid in analyzing the prevalence of diabetes in this population, as well as regional differences within the country. Is the prevalence of undiagnosed diabetes considered high? In addition to comparisons with other countries in the region, the authors could discuss the findings in relation to countries with similar public health service structures.

In the sentence “The study also noted that individuals with primary and secondary education displayed lower awareness and knowledge about diabetes treatment, contributing to a significant link between education and undiagnosed diabetes”, could this low awareness occur for diabetes diagnosis? As so, part of undiagnosed diabetes could be a case of reporting bias? The person could have been diagnosed previously, but not report it appropriately? If you think so, please include it in the text.

In the sentence “This might be due to their ability to consume more glucose, leading to sedentary lifestyles that can contribute to diabetes development”, is there any other explanation? Are the access to health care, glycemic measures, and the diagnosis of diabetes universally distributed in the country? Please explain.

When interpreting the relationship between diabetes and nutritional status, the mechanisms underlying these connections should be more thoroughly explored to aid the readers' understanding. For instance, how do increased technology use and prolonged TV watching influence this relationship, and why is the central region of Bangladesh an example? This part seems unclear to me.

In the limitations section, the authors mention that 'medication history was not included,' yet in the methods, it is stated that those who 'took prescribed medicine' were considered as having diabetes. Does the medication history referenced in the limitations refer to medications other than anti-diabetic drugs? Please clarify.

In the conclusion, when stating that “Overall, the insights gained from this study hold the potential to guide the formulation of targeted interventions aimed at curbing the incidence of undiagnosed diabetes in Bangladesh”, I recommend replace ´incidence´ with ´prevalence´, as the study does not address incidence; Instead, the prevalence of undiagnosed diabetes could be reduced through improved screening.

6. PLOS authors have the option to publish the peer review history of their article (what does this mean? ). If published, this will include your full peer review and any attached files.

**Do you want your identity to be public for this peer review?** For information about this choice, including consent withdrawal, please see our Privacy Policy .

Reviewer #1: No

Reviewer #2: **Yes: ** Betine Pinto Moehlecke Iser

---

## [Author Response · Author response to Decision Letter 0]

7 Oct 2024

Date: October 5, 2024

To

Editor

Plos One

RE: Prevalence and regional disparities of Undiagnosed Diabetes Mellitus in Bangladesh: Results from the Bangladesh Demographic and Health Survey data

Dear Editor:

Thank you very much for your editorial suggestions and the reviewers’ comments. They were accommodating. Please find enclosed an itemized list of responses along with the revised manuscript.

In our response to the reviewer, we used regular font for the comments/questions by the referees and regular, bold font for our responses, which are shown immediately following the questions/comments.

Thank you once again for the opportunity to submit a revised manuscript.

Ahmed Hossain, PhD

Professor, Department of Public Health

Director, Global Health Institute

North South University.

E-mail: ahossain@sharjah.ac.ae

Editor comments:

• The manuscript has potential but needs a deep review at the moment.

Authors: Thank you.

• The introduction needs to be more concise and focused on diabetes. What was the need to mention hypertension? Or the definition of diabetes? My suggestion is to present an overview of data (for the Southeast Asian countries, then Bangladesh), the lack of knowledge and finally the importance of the current study and its objectives.

Authors: Thank you for your suggestion. We rewrote the Introduction section.

• For methods:

o It is crucial to give more detailed information on the Bangladesh Demographic and Health Survey (BDHS).

Authors: Many thanks for your wonderful suggestion. We have included a brief details about the BDHS data.

The Bangladesh Demographic and Health Survey (BDHS) is a comprehensive national survey that gathers extensive data on various health and demographic indicators within Bangladesh [19]. The National Institute of Population Research and Training (NIPORT), under the Ministry of Health and Family Welfare, oversees the periodic conduct of this survey. The data collected is essential for informing health policy decisions and program implementation in Bangladesh. The BDHS is part of the global Demographic and Health Surveys (DHS) program, which is primarily funded by the United States Agency for International Development (USAID) and receives technical support from ICF International. Through its systematic collection of data, the BDHS provides valuable insights that contribute to understanding health trends and challenges in the country.

o What do the authors mean by "We classified the wealth status as a household-level element as Poor, Middle, and Rich groups.". Did you come up with this classification based on what? Is there a reference? If yes, you need to add it.

Authors: In the BDHS, wealth status is assessed using a wealth index that classifies households into quintiles based on assets and amenities, such as ownership of consumer goods and housing characteristics. This index is calculated through principal component analysis (PCA) to rank households into five categories: Poorest, Poorer, Middle, Richer, and Richest. For analysis, the poorest and poorer categories are combined as "poor," while the richer and richest are grouped as "rich," resulting in three overall wealth status categories.

In the BDHS, wealth status is assessed using a wealth index that classifies households into quintiles based on assets and amenities, such as ownership of consumer goods and housing characteristics. This index is calculated through principal component analysis (PCA) to rank households into five categories: Poorest, Poorer, Middle, Richer, and Richest. For analysis, the poorest and poorer categories are combined as "poor," while the richer and richest are grouped as "rich," resulting in three overall wealth status categories.

• The discussion needs an improvement. The authors explored poorly the previous cited study from 2019 which has found a higher prevalence (was it really higher? can't tell since the authors have not presented CI95%), considering diagnosed diabetes. The BDHS was conducted just one year before. Do the authors discussed the diffference?

Authors: Thanks for your suggestion. We rewrote the discussion section according to your suggestion.

• What are the possiblle interventions you mention in Conclusions that could help reduce diabetes in Bangladesh? This conclusion is very broad. Please provide some examples.

Authors: We rewrote the section and the following conclusion has been drawn:

The high prevalence of untreated diabetes remains a significant concern in Bangladesh, contributing to substantial health threats for the population. Overall, the results highlight complex interactions between demographic factors and diabetes prevalence in the studied population. The significant associations found among age, education level, wealth status, hypertension, BMI, and regional disparities underscore the need for tailored public health interventions that address these specific determinants of health. Future research should focus on understanding the underlying mechanisms driving these associations and developing targeted strategies to improve screening, prevention, and management of diabetes across diverse population segments. By addressing these factors holistically, we can enhance health outcomes and reduce the burden of diabetes within communities.

---

## [Decision Letter · Decision Letter 1]

16 Jan 2025

PONE-D-23-27830R1Prevalence and regional disparities of Undiagnosed Diabetes Mellitus in Bangladesh: Results from the Bangladesh Demographic and Health Survey dataPLOS ONE

Dear Dr. Hossain,

Thank you for submitting your manuscript to PLOS ONE. After careful consideration, we feel that it has merit but does not fully meet PLOS ONE’s publication criteria as it currently stands. Therefore, we invite you to submit a revised version of the manuscript that addresses the points raised during the review process.  **Academic Editor's Comments:** **I have thoroughly reviewed your manuscript in conjunction with the detailed feedback provided by two experienced reviewers. After careful consideration, I share their concerns regarding several key aspects of your work. Based on their constructive recommendations, I invite you to revise and resubmit your manuscript for further evaluation. To strengthen your paper, it is essential that you address the specific issues and suggestions highlighted by the reviewers.**

**I look forward to receiving your revised version and seeing how you've incorporated their valuable insights.**

We look forward to receiving your revised manuscript.

Kind regards,

Md Mohsan Khudri, Ph.D.

Academic Editor

PLOS ONE

Reviewers' comments:

Reviewer's Responses to Questions

**Comments to the Author**

1. If the authors have adequately addressed your comments raised in a previous round of review and you feel that this manuscript is now acceptable for publication, you may indicate that here to bypass the “Comments to the Author” section, enter your conflict of interest statement in the “Confidential to Editor” section, and submit your "Accept" recommendation.

Reviewer #3: (No Response)

Reviewer #4: (No Response)

2. Is the manuscript technically sound, and do the data support the conclusions?

Reviewer #3: Partly

Reviewer #4: Partly

3. Has the statistical analysis been performed appropriately and rigorously? 

Reviewer #3: Yes

Reviewer #4: No

4. Have the authors made all data underlying the findings in their manuscript fully available?

Reviewer #3: Yes

Reviewer #4: Yes

5. Is the manuscript presented in an intelligible fashion and written in standard English?

Reviewer #3: Yes

Reviewer #4: Yes

6. Review Comments to the Author

**Reviewer #3: ** Unfortunately, I did not participate in the first round of evaluations, so my observations were not incorporated until this second round, which should contain specific aspects and not be as generic as the following:

Consideration 1:

The RR is usually defined as the incidence rate of the outcome in the treatment or exposed group divided by the incidence rate of the outcome in the control or non-exposed group. The RRR is defined as a risk reduction. These indicators or effect measures are used in longitudinal studies, not in cross-sectional studies.

Consideration 2:

In logistic regression (binomial or multinomial), the coefficients of the logit (β), if they are exponents (expβ), are interpreted as ORs. These ORs are used to measure the association between risk factors and the prevalence of a condition. Although ORs are not exactly the same as prevalence ratios (PRs), they can be interpreted in a similar way to understand the relationship between variables.

For these reasons, it is preferable to use OR rather than RRR.

On other hand, ORs in a prevalence study do not allow the following to be said:

“Education played a significant role: individuals with primary and secondary education had 52% (RRR = 1.52, 95% CI = 1.07-2.15) and 63% (RRR = 1.63, 95% CI = 1.12-2.38) higher risk of undiagnosed diabetes, respectively, compared to those with no formal education”.

However, the following paragraphs are more appropriate:

• Education played a significant role: individuals with primary and secondary education had 52% (OR = 1.52, 95% CI = 1.07-2.15) and 63% (OR = 1.63, 95% CI = 1.12-2.38) increased odds of having undiagnosed diabetes, respectively, compared to those with no formal education.

• The odds of having undiagnosed diabetes were 1.5 times (OR = 1.52, 95% CI = 1.07-2.15) and 1.6 times (OR = 1.63, 95% CI = 1.12-2.38) higher for those with primary and secondary education, respectively, compared with those with no formal education.

• Education played a significant role: individuals with primary and secondary education had 52% (OR = 1.52, 95% CI = 1.07-2.15) and 63% (OR = 1.63, 95% CI = 1.12-2.38) higher prevalence of undiagnosed diabetes, respectively, compared to those with no formal education.

Other paragraphs need to be improved:

“We found hypertension is strongly associated with an increased risk of both diagnosed and undiagnosed diabetes”

This is because the study did not have a follow-up period, and it is not possible to calculate incidence (risk) rates. The following paragraph is more appropriate:

“We found hypertension is strongly associated with both diagnosed and undiagnosed diabetes”

It is recommended that authors read the following document in order to use the terms correctly:

Hsueh, L., Wu, W., Hirsh, A. T., de Groot, M., Mather, K. J., & Stewart, J. C. Undiagnosed diabetes among immigrant and racial/ethnic minority adults in the United States: National Health and Nutrition Examination Survey 2011-2018. Annals of Epidemiology 2020, 51: 14–19. https://doi.org/10.1016/j.annepidem.2020.07.009

Additionally, other limitations must be included. Firstly, glucose levels were measured using capillary blood, and although a conversion to plasma glucose was performed, this method may still introduce some inaccuracies. Second, a single blood glucose measurement is not sufficient to diagnose diabetes. It is possible that some people diagnosed with diabetes may not have the condition on a second measurement of fasting plasma glucose (false positive) and, conversely, some people who do have diabetes may be missed (false negative).

**Reviewer #4:**  While the manuscript addresses a critical and important topic, it requires substantial revisions before it can be considered for publication. My major comments are outlined below:

Methods:

• Compared to the standard multinomial logistic regression model, multilevel multinomial logistic regression analysis offers significant advantages. It reduces parameter overestimation and provides more accurate estimates, particularly in hierarchical data structures like the DHS survey. The authors are encouraged to consider this approach or discuss why they opted for the current method.

• To assess the stability of a multinomial logistic regression model applied to cross-sectional data, various methods focusing on model assumptions, goodness-of-fit, and predictive performance across data subsets can be employed. This assessment is missing in the manuscript. Additionally, the results of multicollinearity checks were not reported. Addressing these aspects would strengthen the validity of the findings.

• The adequacy of the sample size for a stable multinomial logistic regression model is a critical consideration, especially given the number of predictor variable categories. Adhering to the events per predictor parameter (EPP) rule ensures the statistical soundness and reliability of the findings. Harrell (2001, DOI: 10.1007/978-3-319-19425-7) suggests at least 15 EPP, and others recommend at least 20 EPP (e.g., Ogundimu et al., 2016, DOI: 10.1016/j.jclinepi.2016.02.031; Austin & Steyerberg, 2014, DOI: 10.1177/0962280214558972). Based on Table 1, adhering to the 15 EPP rule requires at least 345 events, but the smallest outcome category (i.e., undiagnosed diabetes) has only 304 events. This should be discussed in the limitations section.

• The peer reviewer’s comment on this was not addressed in the revision. The statement, ‘We classified the wealth status as a household-level element as Poor, Middle, and Rich groups,” lacks clarity. Was this classification based on a specific reference or methodology? If so, please provide the source. If it was developed independently, additional justification and details are needed.

• It is unclear whether a specific framework or rationale was followed for selecting predictors in the model. Did the authors rely on theoretical considerations, empirical evidence, or data-driven approaches (e.g., stepwise selection)? If so, please state this clearly and justify the choices made. Providing transparency here would enhance the study’s methodological rigor.

Discussion

• The manuscript’s discussion section restates key findings and compares them to studies in agreement but offers limited mechanistic explanations for the observed patterns. Providing insights into the underlying biological, behavioral, or systemic factors contributing to these patterns would enhance the discussion.

• The manuscript highlights regional differences but does not explore the underlying causes of these disparities in depth. Evidence suggests that people living in coastal areas may experience a higher prevalence of diabetes than those in inland regions due to a combination of lifestyle, dietary habits, and environmental factors (DOI: 10.1136/bmjdrc-2015-000110; 10.1186/1471-2458-13-299; 10.15226/2374-6890/4/3/00181), which is not in line with the study findings. Expanding on these aspects would better align the discussion with the title’s focus.

• Overall, the whole discussion section requires a thorough review. Some examples:

a) The statement, "Interestingly, individuals with college education exhibit a lower risk for diagnosed diabetes but a higher risk for undiagnosed diabetes," should be revised because the RRRs for diagnosed diabetes are not statistically significant.

b) Similarly, for undiagnosed diabetes, terms like “greater” should not be used when the 95% confidence intervals for overweight and obese categories overlap.

c) The claim, “The analysis reveals a difference in diabetes prevalence based on residence type,” should be clarified or revised. While regional differences are evident, no significant differences by residence type (urban, rural, semi-urban) were found.

d) To explore region/community-level factors, incorporating data from the community questionnaire, such as the distance to healthcare facilities, could provide valuable insights into regional disparities.

7. PLOS authors have the option to publish the peer review history of their article (what does this mean? ). If published, this will include your full peer review and any attached files.

**Do you want your identity to be public for this peer review?** For information about this choice, including consent withdrawal, please see our Privacy Policy .

Reviewer #3: No

Reviewer #4: **Yes: ** Karar Zunaid Ahsan

---

## [Author Response · Author response to Decision Letter 1]

12 Feb 2025

Date: February 11, 2025

To

Editor

Plos One

RE: Prevalence and regional disparities of Undiagnosed Diabetes Mellitus in Bangladesh: Results from the Bangladesh Demographic and Health Survey data

PONE-D-23-27830R1

Dear Editor:

Thank you very much for your editorial suggestions and the reviewers’ comments. They were accommodating. Please find enclosed an itemized list of responses along with the revised manuscript.

In our response to the reviewer, we used regular font for the comments/questions by the referees and regular, bold font for our responses, which are shown immediately following the questions/comments.

Thank you once again for the opportunity to submit a revised manuscript.

Ahmed Hossain, PhD

Professor, Health Care Management, University of Sharjah, Sharjah, UAE.

and

Director, Global Health Institute

North South University.

E-mail: ahossain@sharjah.ac.ae

Table of Contents

REVIEWER #3: 2

REVIEWER #4: 4

Reviewer #3:

Unfortunately, I did not participate in the first round of evaluations, so my observations were not incorporated until this second round, which should contain specific aspects and not be as generic as the following:

Consideration 1:

The RR is usually defined as the incidence rate of the outcome in the treatment or exposed group divided by the incidence rate of the outcome in the control or non-exposed group. The RRR is defined as a risk reduction. These indicators or effect measures are used in longitudinal studies, not in cross-sectional studies.

Authors: Multinomial logistic regression is a powerful statistical tool used in cross-sectional studies to model outcomes with more than two categories, allowing researchers to estimate relative risk ratios (RRRs) for each category relative to a baseline category (Camey et al.). The multinomial logit model is often preferred over traditional regression approaches in cross-sectional studies due to its theoretical and empirical advantages (Gensch et al., Dixit et al.). It is crucial to avoid using odds ratio (OR) as a proxy for RRR or PR in multinomial logistic regression, as this can lead to significant misinterpretations (Camey et al. and Blizzard et al.). For multinomial outcomes the OR must not be used as an approximation of the RR or PR, since this may lead to incorrect conclusions (Camey et al.).

Gensch, D., & Recker, W. (1979). The Multinomial, Multiattribute Logit Choice Model. Journal of Marketing Research, 16, 124 - 132. https://doi.org/10.1177/002224377901600117.

Dixit, S., Kumar, B., Singh, A., & Ashoka, R. (2015). An Application of Multinomial Logistic Regression to Assess the Factors Affecting the Women to Be Underweight and Overweight: A Practical Approach. -. International Journal of Health Sciences and Research, 5, 11-17.

Camey, S., Torman, V., Hirakata, V., Cortes, R., & Vigo, Á. (2014). Bias of using odds ratio estimates in multinomial logistic regressions to estimate relative risk or prevalence ratio and alternatives.. Cadernos de saude publica, 30 1, 21-9 . https://doi.org/10.1590/0102-311X00077313.

Blizzard, L., & Hosmer, D. (2006). The Log Multinomial Regression Model for Nominal Outcomes with More than Two Attributes. Biometrical Journal, 49. https://doi.org/10.1002/bimj.200610377.

Consideration 2:

In logistic regression (binomial or multinomial), the coefficients of the logit (β), if they are exponents (expβ), are interpreted as ORs. These ORs are used to measure the association between risk factors and the prevalence of a condition. Although ORs are not exactly the same as prevalence ratios (PRs), they can be interpreted in a similar way to understand the relationship between variables.

For these reasons, it is preferable to use OR rather than RRR.

On other hand, ORs in a prevalence study do not allow the following to be said:

“Education played a significant role: individuals with primary and secondary education had 52% (RRR = 1.52, 95% CI = 1.07-2.15) and 63% (RRR = 1.63, 95% CI = 1.12-2.38) higher risk of undiagnosed diabetes, respectively, compared to those with no formal education”.

However, the following paragraphs are more appropriate:

• Education played a significant role: individuals with primary and secondary education had 52% (OR = 1.52, 95% CI = 1.07-2.15) and 63% (OR = 1.63, 95% CI = 1.12-2.38) increased odds of having undiagnosed diabetes, respectively, compared to those with no formal education.

• The odds of having undiagnosed diabetes were 1.5 times (OR = 1.52, 95% CI = 1.07-2.15) and 1.6 times (OR = 1.63, 95% CI = 1.12-2.38) higher for those with primary and secondary education, respectively, compared with those with no formal education.

• Education played a significant role: individuals with primary and secondary education had 52% (OR = 1.52, 95% CI = 1.07-2.15) and 63% (OR = 1.63, 95% CI = 1.12-2.38) higher prevalence of undiagnosed diabetes, respectively, compared to those with no formal education.

Authors: We sincerely appreciate your valuable suggestion. For consistency and clarity, we have retained the presentation of results as Relative Risk Ratios (RRR) derived from the multinomial logistic regression model.

Other paragraphs need to be improved:

“We found hypertension is strongly associated with an increased risk of both diagnosed and undiagnosed diabetes”

This is because the study did not have a follow-up period, and it is not possible to calculate incidence (risk) rates. The following paragraph is more appropriate:

“We found hypertension is strongly associated with both diagnosed and undiagnosed diabetes”

Authors: Thank you for your wonderful suggestion. We corrected it accordingly.

It is recommended that authors read the following document in order to use the terms correctly:

Hsueh, L., Wu, W., Hirsh, A. T., de Groot, M., Mather, K. J., & Stewart, J. C. Undiagnosed diabetes among immigrant and racial/ethnic minority adults in the United States: National Health and Nutrition Examination Survey 2011-2018. Annals of Epidemiology 2020, 51: 14–19. https://doi.org/10.1016/j.annepidem.2020.07.009

Authors: Many thanks for your reference. The article that you recommended has outcome with two categories, but we have an outcome with three categories.

Additionally, other limitations must be included. Firstly, glucose levels were measured using capillary blood, and although a conversion to plasma glucose was performed, this method may still introduce some inaccuracies. Second, a single blood glucose measurement is not sufficient to diagnose diabetes. It is possible that some people diagnosed with diabetes may not have the condition on a second measurement of fasting plasma glucose (false positive) and, conversely, some people who do have diabetes may be missed (false negative).

Authors: Thanks. The article incorporates the limitations suggested in our discussion.

Glucose levels were measured using capillary blood, and while a conversion to plasma glucose was applied, this approach may still result in some inaccuracies. Moreover, a single blood glucose measurement to diagnose diabetes in the study can lead to diagnostic errors due to its potential for false positives and false negatives. 

Reviewer #4:

While the manuscript addresses a critical and important topic, it requires substantial revisions before it can be considered for publication. My major comments are outlined below:

Methods:

• Compared to the standard multinomial logistic regression model, multilevel multinomial logistic regression analysis offers significant advantages. It reduces parameter overestimation and provides more accurate estimates, particularly in hierarchical data structures like the DHS survey. The authors are encouraged to consider this approach or discuss why they opted for the current method.

Authors: Thank you very much for your wonderful suggestion. We applied the multilevel multinomial logistic regression model and the results are shown in sensitivity analysis.

We conducted a sensitivity analysis using a multilevel multinomial regression model implemented with the brms package in R. The model applies the geographic residence as the random effect and results are presented in the Appendix. The multilevel multinomial logistic regression model was employed to examine regional disparities in the prevalence of diagnosed and undiagnosed diabetes compared to no diabetes. In the results, each parameter is summarized using the mean (Estimate) and standard deviation (Est.Error) of the posterior distribution, along with two-sided 95% Credible Intervals (l-95% CI and u-95% CI) derived from quantiles.

The findings on regional disparities indicate that individuals in city areas are exp(0.60) = 1.822 times more likely to have diabetes compared to those in rural regions. The one-sided 95% credibility interval does not include zero, suggesting statistical significance. Additionally, city areas show that individuals are exp(0.35) = 1.42 times more likely to have undiagnosed diabetes compared to rural regions. Similar results were observed when using a multinomial regression model.

• To assess the stability of a multinomial logistic regression model applied to cross-sectional data, various methods focusing on model assumptions, goodness-of-fit, and predictive performance across data subsets can be employed. This assessment is missing in the manuscript. Additionally, the results of multicollinearity checks were not reported. Addressing these aspects would strengthen the validity of the findings.

Authors: Thank you. We evaluated the validity of the findings and results are shown in Appendix.

The AIC value of 7629.43 suggests that the model achieves a reasonable balance between fit and complexity. Furthermore, when compared to the null model, the reduction in the AIC value indicates that this model provides a better fit to the data.

The residual deviance of 7577.43 also points to a good fit of the model to the data. Its significance can be further assessed by comparing it to the null deviance, which helps determine the extent to which the model improves over a baseline model with no predictors.

Null model:

• The adequacy of the sample size for a stable multinomial logistic regression model is a critical consideration, especially given the number of predictor variable categories. Adhering to the events per predictor parameter (EPP) rule ensures the statistical soundness and reliability of the findings. Harrell (2001, DOI: 10.1007/978-3-319-19425-7) suggests at least 15 EPP, and others recommend at least 20 EPP (e.g., Ogundimu et al., 2016, DOI: 10.1016/j.jclinepi.2016.02.031; Austin & Steyerberg, 2014, DOI: 10.1177/0962280214558972). Based on Table 1, adhering to the 15 EPP rule requires at least 345 events, but the smallest outcome category (i.e., diagnosed diabetes) has only 333 events. This should be discussed in the limitations section.

Authors: Thank you for your wonderful suggestion. We added the following statement in the limitation:

The Events Per Predictor (EPP) rule is a widely accepted guideline used to determine the minimum number of events (observations) needed for each category of outcome variable to ensure the reliability of a multinomial logistic regression model. In this case, the sample size for the smallest outcome category (diagnosed diabetes, with 333 events) falls short of the recommended EPP threshold of 15, which would require at least 345 events. This discrepancy highlights a potential limitation in the model's stability and reliability.

• The peer reviewer’s comment on this was not addressed in the revision. The statement, ‘We classified the wealth status as a household-level element as Poor, Middle, and Rich groups,” lacks clarity. Was this classification based on a specific reference or methodology? If so, please provide the source. If it was developed independently, additional justification and details are needed.

Authors: Thank you. It is briefly explained in the manuscript with a reference.

In the BDHS, wealth status is assessed using a wealth index that classifies households into quintiles based on assets and amenities, such as ownership of consumer goods and housing characteristics [20]. This index is calculated through principal component analysis (PCA) to rank households into five categories: Poorest, Poorer, Middle, Richer, and Richest. For analysis, the poorest and poorer categories are combined as "poor," while the richer and richest are grouped as "rich," resulting in three overall wealth status categories.

USAID The DHS Program: Wealth Index Available from: https://dhsprogram.com/topics/wealth-index/

• It is unclear whether a specific framework or rationale was followed for selecting predictors in the model. Did the authors rely on theoretical considerations, empirical evidence, or data-driven approaches (e.g., stepwise selection)? If so, please state this clearly and justify the choices made. Providing transparency here would enhance the study’s methodological rigor.

Authors: We previously utilized a directed acyclic graph (DAG) – a visual representation of hypothesized causal relationships – to identify confounders in these relationships and the plot is given in the Appendix. In this DAG (detailed in Supplement, constructed with DAGitty: http://www.dagitty.net/dags.html#), diabetes status was considered the outcome, while residency was the main exposure. For the association between exposures and diabetes, a minimal sufficient adjustment set including age, sex, BMI, hypertension and residency was determined.

Discussion

• The manuscript’s discussion section restates key findings and compares them to studies in agreement but offers limited mechanistic explanations for the observed patterns. Providing insights into the underlying biological, behavioral, or systemic factors contributing to these patterns would enhance the discussion.

Authors: The discussion section is rewritten.

• The manuscript highlights regional differences but does not explore the underlying causes of these disparities in depth. Evidence suggests that people living in coastal areas may experience a higher prevalence of diabetes than those in inland regions due to a combination of lifestyle, dietary habits, and environmental factors (DOI: 10.1136/bmjdrc-2015-000110; 10.1186/1471-2458-13-299; 10.15226/2374-6890/4/3/00181), which is not in line with the study findings. Expanding on these aspects would better align the discussion with the title’s focus.

Authors: Thank you for your insightful comment. We have addressed this by including mosaic plots in the Appendix, and the corresponding findings are discussed in detail within the manuscript.

The plots depict the distribution of various demographic, health, and socioeconomic factors across residential regions (City, Rural, and Semi-Urban). Key observations include:

• Older populations (60+ years) are more concentrated in rural and semi-urban areas, likely due to aging in place and the migration of younger generations to urban centers.

• Urban areas exhibit a higher prevalence of hypertension, potentially driven by lifestyle factors such as consumption of processed foods, physical inactivity, and stress.

• The prevalence of obesity and overweight individuals is greater in urban regions, likely influenced by dietary habits, sedentary lifestyles, and socioeconomic status.

• In contrast, underweight individuals are more common in rural areas, possibly linked to lower income levels and nutritional deficiencies.

Regarding education:

• Higher education levels (college or above) are significantly associated with urban areas, likely due to better access to universities and educational institutions.

• Primary or no education is more prevalent in rural regions, highlighting disparities in access to educational resources.

• In terms of employment and income, urban areas have higher employment rates, reflecting greater job availability in industries, services, and corporate sectors.

• Rural areas show a higher proportion of individuals who are "not working," which may include agricultural laborers or unemployed individuals.

• Urban residents are more likely to fall into the "rich" category, reflecting better income opportunities and higher living costs.

• Rural areas have a higher percentage of "poor" individuals, consistent with l

---

## [Decision Letter · Decision Letter 2]

3 Mar 2025

Prevalence and regional disparities of Undiagnosed Diabetes Mellitus in Bangladesh: Results from the Bangladesh Demographic and Health Survey data

PONE-D-23-27830R2

Dear Dr. Hossain,

We’re pleased to inform you that your manuscript has been judged scientifically suitable for publication and will be formally accepted for publication once it meets all outstanding technical requirements.

Kind regards,

Md Mohsan Khudri, Ph.D.

Academic Editor

PLOS ONE

Additional Editor Comments (optional):

Reviewers' comments:

Reviewer's Responses to Questions

**Comments to the Author**

1. If the authors have adequately addressed your comments raised in a previous round of review and you feel that this manuscript is now acceptable for publication, you may indicate that here to bypass the “Comments to the Author” section, enter your conflict of interest statement in the “Confidential to Editor” section, and submit your "Accept" recommendation.

Reviewer #3: All comments have been addressed

Reviewer #4: All comments have been addressed

2. Is the manuscript technically sound, and do the data support the conclusions?

Reviewer #3: Yes

Reviewer #4: Yes

3. Has the statistical analysis been performed appropriately and rigorously? 

Reviewer #3: Yes

Reviewer #4: Yes

4. Have the authors made all data underlying the findings in their manuscript fully available?

Reviewer #3: Yes

Reviewer #4: Yes

5. Is the manuscript presented in an intelligible fashion and written in standard English?

Reviewer #3: Yes

Reviewer #4: Yes

6. Review Comments to the Author

Reviewer #3: The authors have adequately responded to all the issues raised. They have provided bibliography that clarifies the points raised in my previous report.

Reviewer #4: I thank the authors for thoughtfully incorporating my peer review comments into their manuscript. I believe the revisions have strengthened the clarity and impact of the manuscript.

For the authors' information, the statement that "... the dataset does not include information on the distance to healthcare facilities" is not correct. The distance to health facility data are indeed available in the DHS community questionnaire, which can be linked with the DHS individual recode data (at the cluster level) if required.

Again, I appreciate the authors' careful attention to the feedback and commend their efforts to improve the manuscript.

7. PLOS authors have the option to publish the peer review history of their article (what does this mean? ). If published, this will include your full peer review and any attached files.

**Do you want your identity to be public for this peer review?** For information about this choice, including consent withdrawal, please see our Privacy Policy .

Reviewer #3: No

Reviewer #4: **Yes: ** Karar Zunaid Ahsan, PhD

---

## [Editor Report · Acceptance letter]

PONE-D-23-27830R2

PLOS ONE

Dear Dr. Hossain,

I'm pleased to inform you that your manuscript has been deemed suitable for publication in PLOS ONE. Congratulations! Your manuscript is now being handed over to our production team.

Kind regards,

on behalf of

Dr. Md Mohsan Khudri

Academic Editor

PLOS ONE